# ICU ‘Magic Numbers’: The Role of Biomarkers in Supporting Clinical Decision-Making

**DOI:** 10.3390/diagnostics15080975

**Published:** 2025-04-11

**Authors:** Francesco Cipulli, Eleonora Balzani, Giuseppe Marini, Sergio Lassola, Silvia De Rosa, Giacomo Bellani

**Affiliations:** 1Anesthesia and Intensive Care 1, Santa Chiara Hospital, Azienda Provinciale per i Servizi Sanitari (APSS), Largo Medaglie d’Oro 9, 38112 Trento, Italy; giuseppe.marini@apss.tn.it (G.M.); sergio.lassola@apss.tn.it (S.L.); silvia.derosa@apss.tn.it (S.D.R.); giacomo.bellani@apss.tn.it (G.B.); 2Centre for Medical Sciences-CISMed, University of Trento, Via S. Maria Maddalena 1, 38122 Trento, Italy; eleonora.balzani@apss.tn.it

**Keywords:** ICU biomarkers, procalcitonin, NT-proBNP, prealbumin, interleukin-6, neuron-specific enolase, serum creatinine, cystatin C, activated clotting time

## Abstract

Critical care medicine is a highly complex field where diagnosing diseases and selecting effective therapies pose daily challenges for clinicians. In critically ill patients, biomarkers can play a crucial role in identifying and addressing clinical problems. Selecting the right biomarkers and utilizing them effectively can lead to more informed decisions, ultimately impacting patient outcomes. However, each biomarker has its strengths and limitations, making a thorough understanding essential for accurate diagnosis and treatment management. For instance, neuron-specific enolase (NSE) is commonly used to predict outcomes in out-of-hospital cardiac arrest (OHCA), procalcitonin (PCT) levels strongly correlate with bacterial infections, and NT-proBNP serves as a reliable indicator of cardiac stress. Additionally, serum creatinine (SCr) remains fundamental in renal diagnostics, while prealbumin helps differentiate catabolic and anabolic phases in critically ill patients. This narrative review highlights a carefully selected set of biomarkers known for their clinical utility and reliability in guiding critical care decisions. Further refining the application of biomarkers—especially by integrating them into a multimodal approach—will enhance clinicians’ ability to navigate the challenges of critical care, always striving to improve patient outcomes.

## 1. Introduction

Critical care medicine is a complex activity where knowledge, experience, intuition, and sometimes even luck converge. Diagnosing diseases and selecting effective therapies represent a daily challenge for clinicians, requiring them to analyze and process vast amounts of information. Patient’s medical history, clinical signs and symptoms, laboratory results, and radiological findings are just some of the elements that must be considered during routine practice. At times, the sheer volume of data can overwhelm doctors, exposing them to the risk of becoming lost in a maze of irrelevant information. This narrative review focuses on a curated set of biomarkers renowned for their reliability in supporting clinical decision-making. By leveraging these tools, we aim to streamline decision-making processes and enhance patient outcomes in the ICU’s high-stakes environment.

## 2. Neuron-Specific Enolase (NSE) as a Prognostic Marker of Central Nervous System Damage and Poor Neurological Outcomes Following Cardiac Arrest

NSE is an enzyme released during neuronal cell necrosis or apoptosis, serving as a biomarker to stratify illness severity and predict outcomes in various conditions, including sepsis-associated encephalopathy (SAE) [1], status epilepticus (SE) [2], and out-of-hospital cardiac arrest (OHCA) [3]. At the molecular level, neuron-specific enolase (NSE) is the γ-isoform of the glycolytic enzyme enolase, encoded by the ENO2 gene. During neuronal maturation, this isoenzyme progressively replaces Non-Neuronal Enolase (NNE), encoded by the ENO1 gene, marking a critical transition that reflects neuronal differentiation and specialization [4]. The predominance of NSE in mature neurons enhances their metabolic efficiency, supporting the high energy demands required for functions such as synaptic transmission and cellular signaling [5]. NSE serves as a critical biomarker in various neurological conditions, providing insights into disease severity and aiding diagnostic accuracy. In sepsis-associated encephalopathy (SAE), elevated NSE levels have been shown to correlate with disease severity. A meta-analysis of 682 SAE patients and 946 controls revealed a significant mean difference of 7.79 ng/mL (95% CI: 5.23–10.34, I^2^ = 99%) between groups [1]. Furthermore, a cut-off value of 14.36 μg/L on day 3 demonstrated moderate diagnostic accuracy, with a sensitivity of 61.1% and specificity of 73.9% [6]. In status epilepticus (SE), NSE’s diagnostic utility improves when used alongside complementary biomarkers. Individually, NSE exhibited limited accuracy (AUC: 0.624, sensitivity: 77.3%, specificity: 45.2%). However, its combination with S100β, a biomarker of astrocytic damage, significantly enhanced diagnostic performance, achieving an AUC of 0.748 [2]. In out-of-hospital cardiac arrest (OHCA), NSE levels measured at specific time points provide critical prognostic insights. Serum NSE levels exceeding 20 μg/L on days 3–4 post-arrest are strongly associated with poor neurological outcomes, with a sensitivity of 85% and specificity of 82%. Levels surpassing 30 μg/L on day 4 offer 100% specificity for predicting poor outcomes [3]. Specificity improves progressively over time, increasing from 70 to 75% on day 1 to 85–90% by days 3–4, as NSE accumulates due to sustained neuronal injury and disruption of the blood–brain barrier [7]. Serial NSE measurements further refine prognostic accuracy. For instance, the 48:24 h NSE ratio enhances predictive precision, with a ratio ≥1.7 achieving 100% specificity for poor outcomes [8]. These findings highlight the utility of time-dependent NSE measurements and ratios for reliable outcome prediction in OHCA patients. While NSE is a valuable biomarker, its diagnostic reliability is significantly enhanced when integrated with other tools. Somatosensory evoked potentials (SSEPs), which evaluate cortical responses to peripheral nerve stimulation, are highly specific for predicting poor neurological outcomes. The absence of the N20 response is nearly 100% specific, though its sensitivity is limited, ranging from 30 to 60% [9]. Electroencephalography (EEG), particularly when performed within the first 24 h, identifies malignant patterns such as burst suppression and isoelectric activity, with specificity levels between 90 and 100% [9]. Neuroimaging, including CT and MRI, complements functional assessments. CT findings, such as a reduced gray-to-white matter ratio, exhibit a specificity of 98%, while MRI, particularly diffusion-weighted imaging, can detect subtle anoxic injuries with high sensitivity [7]. Clinical exams remain indispensable in prognostication. The absence of brainstem reflexes and a motor Glasgow Coma Scale (GCS) score of ≤2 at 72 h are strongly predictive of poor outcomes [9]. Despite its utility, NSE levels are influenced by confounding factors such as hemolysis, which can artificially elevate values by up to 50% [10]. Therefore, while NSE is a dynamic and reliable marker, its interpretation must be contextualized within a multimodal prognostic framework, integrating clinical, electrophysiological, and imaging data. Combining neuron-specific enolase (NSE) with complementary biomarkers such as neurofilament light (NfL) or S100β significantly enhances diagnostic accuracy and provides a more comprehensive understanding of neuronal injury. This multimodal approach aids in critical decisions, including escalation or withdrawal of care, ensuring a patient-centered and tailored management strategy [11]. NSE’s utility as a biomarker is best realized when integrated into a broader framework that includes tools such as somatosensory evoked potentials (SSEPs), Electroencephalography (EEG), neuroimaging, and clinical assessments. This robust combination yields precise prognostic insights, empowering clinicians to make informed decisions and deliver individualized care in critical neurological conditions. Ongoing research focused on optimizing NSE thresholds, exploring its dynamics over time, and refining its combination with other biomarkers is paving the way for improved neuro-prognostication strategies. These advancements further solidify NSE’s role as a cornerstone in neurocritical care, facilitating better outcomes for patients with severe neurological injuries (Table 1).

## 3. Procalcitonin (PCT): A Biomarker for Detecting Infections and Guiding Antibiotic Therapy

PCT is a peptide prohormone composed of 116 amino acids and serves as the precursor of calcitonin, a key regulator of calcium homeostasis [13]. Under normal conditions, PCT is produced in the C-cells of the thyroid gland, where it is subsequently cleaved into active calcitonin in response to elevated calcium levels. Calcitonin lowers plasma calcium levels by counteracting parathyroid hormone (PTH) activity, reducing calcium resorption from bone, and promoting renal calcium excretion [14]. In healthy, uninfected individuals, plasma PCT levels remain below 0.5 ng/mL, while calcitonin levels are approximately 10 pg/mL [15]. During bacterial infections, however, non-endocrine tissues such as adipocytes release large quantities of PCT into the bloodstream in response to immuno-inflammatory stimuli, including lipopolysaccharides (LPS), bacterial toxins, Interleukin-6 (IL-6), and Tumor Necrosis Factor α (TNF- α). Plasma PCT levels can rise dramatically, ranging from 1 to 2000 ng/mL, depending on the severity and spread of the infection [16]. Notably, during these infections, plasma calcium levels remain unaffected because the procalcitonin produced by non-endocrine tissues is not converted into active calcitonin. While the exact function of PCT as an acute-phase protein is not yet fully understood, its levels are strongly correlated with bacterial infections, making it a valuable biomarker for differentiating sepsis from other inflammatory conditions.

The 2010 PRORATA trial significantly influenced the clinical use of PCT. This multicenter, prospective, parallel, open-label trial randomized patients into two groups: a control group where antibiotic therapy initiation and discontinuation followed existing guidelines, and an intervention group where these decisions were guided by PCT levels. In the intervention group, PCT levels above 1 ng/mL recommended antibiotic initiation, while levels below 0.5 ng/mL or reductions to less than 20% of the PCT peak supported discontinuation. Although 28-day and 60-day mortality rates were similar in both groups, the PCT-guided group experienced significantly shorter antibiotic exposure. The authors concluded that PCT-guided therapy safely reduces antibiotic exposure and selective pressure without compromising patient survival [17]. PCT demonstrates high sensitivity and specificity, with false positives and false negatives being relatively rare. False positives can occur following major surgery, trauma, burns, or cardiac arrest, while false negatives are more likely in compartmentalized or early-stage infections. Despite its utility, several aspects of PCT kinetics remain unclear [18]. For instance, acute kidney injury may influence PCT elimination, though current data are limited [19]. Thus, while PCT is a valuable diagnostic and therapeutic tool, it should not be used in isolation. Clinical decisions regarding infection diagnosis and antibiotic strategies should integrate PCT levels within a broader clinical context to ensure accuracy and optimal patient care. (Figure 1).

### Emerging Biomarkers and Future Perspectives for Sepsis Detection: Presepsin

Presepsin is a novel biomarker that has gained attention for its potential role in the early diagnosis and prognosis of sepsis. It is a soluble fragment of the CD14 receptor, which is released into circulation upon activation of the innate immune response to bacterial infections. While it has advantages over traditional markers, as a faster response to the infection, further research is needed to establish standardized cutoff values and optimize its clinical utility in different patient populations [20].

## 4. N-Terminal Pro-Brain Natriuretic Peptide (NT-proBNP): A Reliable Biomarker for Cardiac Failure Diagnosis and Management

NT-proBNP is a 76-amino-acid peptide derived from the cleavage of the proBNP precursor, which is primarily secreted by cardiomyocytes in response to increased wall stress or hemodynamic overload. Upon secretion, proBNP splits into two fragments: BNP (biologically active) and NT-proBNP (biologically inactive), which is then predominantly cleared by the kidneys [21]. Although NT-proBNP itself has no direct biological function, its release closely reflects the pathophysiological processes of the heart [22]. ProBNP is synthesized in response to ventricular wall stretching, often due to volume or pressure overload, and contributes to hemodynamic regulation by inducing vasodilation, lowering blood pressure, and promoting natriuresis and diuresis [21]. NT-proBNP is inactive, but being endowed with greater stability and longer half-life compared to BNP, can serve as a reliable, indirect indicator of cardiac stress (BNP release), characteristics which make it a sensitive and specific biomarker widely recognized for both acute and chronic heart failure [22]. Nevertheless, its high sensitivity and specificity, some clinical conditions can elevate NT-proBNP independently from heart failure. Plasma levels of NT-proBNP can be influenced by several factors such as age, sex, body mass index, chronic renal insufficiency, atrial fibrillation, and certain chronic pulmonary diseases [23]. False positives can also occur in cases of sepsis and pulmonary hypertension, while false negatives are common in obese individuals, probably due to increased peptide distribution volume. Interpreting NT-proBNP levels requires careful consideration of the clinical context and patient-specific factors to avoid diagnostic errors [24]. Elevated NT-proBNP levels are instrumental in confirming heart failure in patients presenting with symptoms such as dyspnea, fatigue, or peripheral edema. The biomarker is particularly helpful in distinguishing cardiac-origin dyspnea from pulmonary-origin dyspnea in emergency settings [25]. Beyond diagnosis, NT-proBNP is a valuable tool for monitoring heart failure progression and assessing treatment response. Reductions in NT-proBNP levels often correlate with clinical improvement, while increases may signal disease worsening or inadequate therapy [26]. Prognostically, elevated NT-proBNP levels are linked to a higher risk of adverse cardiovascular events, including hospitalization and mortality [22]. As such, NT-proBNP also plays a role in risk stratification for patients with cardiovascular diseases. In addition to its central role in heart failure management, NT-proBNP has applications in other clinical contexts, such as perioperative risk assessment and the diagnosis of pulmonary hypertension [27]. NT-proBNP levels can also assist in evaluating volume overload during fluid therapy in critically ill patients [28]. Furthermore, elevated NT-proBNP may indicate conditions such as septic cardiomyopathy or stress-induced cardiomyopathy (SCMP) [29]. However, its use in these contexts should always be complemented by other clinical and diagnostic data to ensure a thorough evaluation of the patient’s condition (Figure 2).

### Emerging Biomarkers and Future Perspectives for Heart Failure: Endotelin-1

Endothelin-1 (ET-1) is a potent vasoconstrictor and a key marker of endothelial dysfunction, playing a critical role in cardiovascular homeostasis, inflammation, and organ perfusion. Its elevated levels in critically ill patients are associated with septic shock, acute heart failure, and pulmonary hypertension, making it a potentially useful biomarker in ICU management. Including Endothelin-1 alongside BNP could enhance ICU hemodynamic monitoring, particularly in patients with sepsis, ARDS, or acute heart failure. Their combined role in assessing vascular function, myocardial stress, and volume status may help optimize vasopressor therapy, fluid resuscitation, and organ protection strategies in critically ill patients [30].

## 5. Interleukin-6 (IL-6): A Key Biomarker of Systemic and Pulmonary Inflammation

IL-6 is a cytokine and an essential biomarker extensively present in numerous acute and chronic inflammatory conditions. It is secreted by a wide range of cells, including T-cells, macrophages, endothelial cells, and fibroblasts, in response to infections, tissue injury, and other pro-inflammatory stimuli [31]. IL-6 plays a dual role, acting as both a pro-inflammatory and anti-inflammatory mediator depending on the physiological context. From a molecular perspective, IL-6 is a glycoprotein consisting of 184 amino acids with a molecular weight of approximately 26 kDa. It signals through the IL-6 receptor complex, which comprises a specific IL-6 binding receptor and the gp130 signal-transducing subunit [32]. Upon receptor binding, IL-6 activates downstream pathways such as JAK/STAT and MAPK, driving cellular responses related to inflammation and immune regulation [33]. IL-6 is critical in the immune response, contributing to the acute-phase reaction by promoting the production of acute-phase proteins such as C-reactive protein (CRP) and fibrinogen in the liver [34]. It is also a key mediator of fever and neutrophil mobilization during infections. In critically ill patients, IL-6 levels correlate with the severity of systemic inflammation and tissue damage, making it a valuable biomarker for prognosis and disease monitoring [33]. Recent studies, including those by Kobayashi et al., highlight the role of IL-6 in Acute Respiratory Distress Syndrome (ARDS) following sepsis [31]. Alveolar macrophages have been shown to significantly express IL-6, reinforcing its role in lung injury-associated inflammation. Matthay et al. further identified IL-6 as a biomarker distinguishing severe sepsis patients with ARDS, emphasizing its diagnostic and prognostic relevance in critical care [33]. IL-6 is a highly sensitive biomarker for systemic inflammation, particularly in sepsis and ARDS, often rising before clinical deterioration [35,36]. However, its specificity is limited, as IL-6 levels can increase in various conditions such as trauma, burns, surgery, or autoimmune diseases [34]. False positives can complicate diagnosis in these scenarios, while delayed or low IL-6 production may occur in immunocompromised patients, potentially masking the severity of inflammation [37]. In intensive care, IL-6 is used to monitor critically ill patients, particularly for early sepsis detection. Elevated IL-6 levels often precede other clinical signs and are associated with the severity of sepsis and mortality risk [37]. In ARDS, IL-6 levels correlate with lung inflammation and tissue damage, reflecting disease progression. During the COVID-19 pandemic, IL-6 gained prominence as a marker of cytokine storm, aiding in identifying patients at risk of rapid deterioration and guiding immunomodulatory therapies like tocilizumab [38]. IL-6 also serves as an indicator of therapeutic response. Reductions in IL-6 levels may suggest effective treatment with anti-inflammatory agents such as corticosteroids or IL-6 receptor antagonists. Additionally, IL-6 has potential applications in stratifying patients for targeted interventions [39]. Despite its clinical utility, interpreting IL-6 levels requires consideration of the broader clinical context due to its non-specificity. Combining IL-6 measurements with other biomarkers like PCT and CRP enhances diagnostic accuracy, particularly for conditions such as ARDS and sepsis [40]. IL-6 is a versatile biomarker with applications in diagnosing and managing inflammatory and critical conditions. Its ability to provide early indications of systemic inflammation and its correlation with disease severity make it invaluable in intensive care settings. However, its interpretation should always be integrated with clinical assessments and additional biomarkers to maximize its diagnostic and prognostic potential.

## 6. Serum Creatinine (SCr) and Cystatin C (CysC): Essential Biomarkers for Renal Function Evaluation

SCr and Cystatin C CysC are essential biomarkers in renal diagnostics, each offering distinct advantages in sensitivity and specificity. SCr, a byproduct of muscle metabolism, is widely used because of its affordability and accessibility. However, its levels are influenced by extrarenal factors such as muscle mass, diet, and age, which can limit its reliability, especially in vulnerable populations like ICU patients [41,42]. On the other hand, CysC, a low-molecular-weight protein produced consistently by all nucleated cells, is emerging as a robust alternative. Its serum concentration, determined primarily by glomerular filtration, makes it a precise marker for estimating the glomerular filtration rate (GFR) and detecting early kidney dysfunction [43,44]. SCr is produced through creatine phosphate metabolism in skeletal muscles, with its levels varying significantly based on factors such as diet and body composition. These variations can compromise its reliability in evaluating kidney function, particularly in critically ill patients [42]. In contrast, CysC is consistently produced and freely filtered by the kidneys, undergoing complete catabolism in the proximal tubules [45,46]. This characteristic makes it a more stable biomarker that is less affected by external variables. While SCr reflects renal function through its elimination via glomerular filtration, it rises slowly in cases of kidney dysfunction. This delay limits its usefulness in detecting early acute kidney injury (AKI), particularly in conditions such as sarcopenia, where muscle mass is reduced [47]. In contrast, CysC, being independent of muscle mass, offers a more accurate and timely assessment of GFR changes, responding faster to AKI or therapeutic interventions [48]. SCr also has low sensitivity for detecting early AKI, often requiring significant GFR reductions (greater than 50%) before its levels noticeably rise. This delayed response restricts its utility for rapid diagnosis [49]. On the other hand, CysC demonstrates superior sensitivity and specificity, with an area under the curve (AUC) of 0.89 for AKI detection. Additionally, it offers greater prognostic value for chronic kidney disease (CKD) progression compared to SCr [50,51]. Despite its limitations, SCr remains the standard biomarker for routine monitoring in stable CKD patients due to its low cost and widespread availability. However, its accuracy is compromised in critically ill patients or those with complicating factors such as malnutrition or fluid overload [52]. Conversely, CysC is particularly effective in acute settings like the ICU, excelling in early AKI detection and renal recovery monitoring. Furthermore, it is a superior predictor of CKD progression and cardiovascular events. The combination of SCr and CysC enhances diagnostic precision, enabling better risk stratification and more personalized treatment plans [53,54]. While SCr remains a cost-effective and accessible tool for routine renal function assessment, its reliability diminishes in complex clinical scenarios due to delayed responses and extrarenal influences. CysC, with its higher sensitivity, specificity, and independence from confounding factors, is invaluable for acute settings such as AKI diagnosis and CKD progression monitoring. By integrating both biomarkers, clinicians can adopt a comprehensive diagnostic approach that improves accuracy and supports tailored clinical interventions in both acute and chronic scenarios (Table 2).

### Emerging Biomarkers and Future Perspectives for AKI: NGAL

Neutrophil gelatinase-associated lipocalin (NGAL) is a biomarker of kidney injury, particularly useful in the early detection of acute kidney injury (AKI). Unlike traditional markers such as serum creatinine (SCr), which rise late during AKI, NGAL levels increase rapidly following renal tubular damage, making it a promising early indicator of kidney dysfunction in critically ill patients. NGAL is released from damaged tubular epithelial cells in response to injury. It is detectable in both plasma and urine within 2–6 h of renal insult, significantly earlier than SCr. Although promising, cutoff values for NGAL are not yet standardized across different clinical settings. Combinations with other biomarkers (e.g., cys-C) may enhance diagnostic accuracy [55].

## 7. Activated Clotting Time (ACT): A Rapid and Reliable Marker for Monitoring Anticoagulation Therapy

ACT is a point-of-care test used to assess whole blood coagulation. It measures the time required for clot formation after activation of the intrinsic pathway, typically induced by celite, kaolin, or similar compounds. The ACT result is expressed in seconds, representing the time taken for the clot to form [56]. Detection methods vary and include monitoring changes in oscillation, resistance to plunger movement (as in Medtronic systems), light absorption, or thrombin exhaustion. The ACT is commonly used to assess the anticoagulant effects of heparin at the bedside, offering a significantly faster turnaround time compared to standard laboratory tests. Unlike activated partial thromboplastin time (aPTT), which becomes unreliable at high heparin concentrations, the ACT exhibits a more linear dose–response curve at the levels required for procedures such as cardiopulmonary bypass, hemodialysis, cardiac catheterization, and vascular surgery [57,58]. However, it is less reliable at lower heparin concentrations [59]. It is important to note that different ACT measurement devices are not interchangeable. Internal validation may be necessary to establish consistency between various machines and cartridges. This validation often requires comparison with a gold-standard reference method, such as the anti-Xa heparin concentration assay, to ensure accuracy and reliability [60]. ACT testing is susceptible to alterations caused by various factors, including temperature fluctuations, hemodilution, the presence of lupus anticoagulant, and platelet function abnormalities [61]. As a result, while the test may initially perform reliably after a heparin bolus—typically achieving target values of 400–500 s—the reliability of ACT can diminish during and towards the end of cardiopulmonary bypass, especially in prolonged procedures. This decline in correlation is particularly notable when comparing ACT values with anti-Xa levels (ranging from 0.3 to 3.4 U/mL) [62]. The weakened correlation can potentially lead to subtherapeutic heparin dosing, underscoring the need for caution and supplementary monitoring in such scenarios. Different ACT devices have been evaluated during hemodialysis, demonstrating good performance for heparin concentrations between 0.2 and 3.3 U/mL, with ACT targets typically ranging from 150 to 200 s, depending on the manufacturer’s reference. The coefficient of variation for this application was found to be between 3 and 5% when compared to anti-Xa levels [63]. However, less reproducible results were observed with systems utilizing glass surfaces for clot activation. This limitation arises from the relatively low heparin concentrations in hemodialysis, which fall near the lower detection range of ACT testing. For other applications involving low heparin dosages, such as the treatment of deep vein thrombosis, ACT testing was evaluated against anti-factor Xa levels (targeting 0.3–0.7 U/mL). In these scenarios, laboratory aPTT showed better reliability and correlation with heparin concentrations, with correlation coefficients of r = 0.72 for ACT and r = 0.74–0.86 for aPTT. Notably, decisions based on ACT agreed with aPTT results only 60% of the time, highlighting the limitations of ACT in these settings [64]. Originally described in 1966, ACT remains a practical and user-friendly option for rapid, repeatable bedside assessment of heparinization. However, interpreting ACT results requires caution, particularly in contexts such as prolonged extracorporeal circulation, residual heparinization, or low heparin dosages. When feasible, laboratory assays should be considered as a complementary or confirmatory method. A deep understanding of the specific ACT system in use is crucial, including awareness of the need for internal validation of cutoff values. For example, ACT targets of 400–450 s for cardiopulmonary bypass and 150–200 s for continuous renal replacement therapy should be confirmed and optimized based on the device and clinical context.

## 8. Prealbumin: A Marker for Differentiating Catabolic and Anabolic Phases in Critically Ill Patients

Prealbumin, also known as transthyretin (TTR), derives its original name from its electrophoretic properties, as it migrates just ahead of the albumin band during electrophoresis. However, despite the similarity in naming, prealbumin is not involved in albumin synthesis [65]. It is a 55 kDa homotetrameric protein primarily synthesized in the liver. Its main function includes serving as a transporter for thyroid hormones, thyroxine (T4) and triiodothyronine (T3), as well as holo-retinol-binding protein (holo-RBP) [66]. TTR has a relatively short half-life of approximately 2.5 days, making it a dynamic marker for changes in protein metabolism. In clinical practice, TTR has been recognized for over 50 years as a marker of nutritional status, but this view has evolved over time. Recent publications have highlighted that its role as a simple indicator of nutritional status could be an oversimplification [67]. Understanding its behavior during the acute phase of illness reveals that it is not merely a passive reactant but an active component of the acute response. TTR production is suppressed by the transcriptional influence of IL-6, and this suppression triggers the release of its bound hormones, an event that leads to a transient hyperthyroid state. This hormonal surge acts as an acute metabolic booster, amplifying the primary cytokine wave and supporting energy demands by promoting lipid oxidation and glycolysis in tissues [68]. Monitoring TTR levels can help clinicians identify high-risk patients who are undergoing severe acute catabolic phases or those with chronic protein–energy malnutrition. By tracking these fluctuations, clinicians can better assess the appropriate timing for nutritional interventions, avoiding the pitfalls of overfeeding, which can worsen patient outcomes [69]. More than just a guide for adjusting caloric or protein intake, TTR levels can help pinpoint the resolution phase, when the patient becomes more nutrient-receptive and enters an anabolic state, signaling a critical window for effective nutritional support [70]. TTR has proven useful in predicting outcomes for patients requiring hemodialysis, particularly at the onset of uremia. Levels below 10 mg/dL compared to those between 10 and 17 mg/dL were associated with a longer hospital stay, with an average of 22 days versus 6 days, respectively [71]. TTR has also been shown to be effective in assessing prognosis in acute heart failure (AHF) patients. A study conducted in a cardiac ICU found that levels below 14 mg/dL at admission and below 15 mg/dL at discharge were strongly correlated with all-cause mortality and composite endpoints of all-cause death or readmission [72]. Similarly, an observational study revealed that AHF patients who died in the hospital had a mean TTR level of 13 mg/dL at admission, compared to 18 mg/dL in those discharged alive (*p* < 0.001), demonstrating a significant predictive value with these cutoffs [73]. Furthermore, in patients with acute coronary disease, a high CRP/TTR ratio was associated with an increased risk of major cardiac complications, with an odds ratio of 1.3 (95% CI: 1.0–1.6). TTR levels of 19.84 ± 3.64 were linked to a higher risk compared to 15.73 ± 4.39 (*p* < 0.001), indicating the importance of monitoring these values in this patient group as well [74]. In the general ICU population, TTR remains a valuable prognostic marker. A 2008 study, conducted by Devakonda et al., found that the mean TTR value on day 1 was strongly associated with survival outcomes, with levels of 14.3 mg/dL in survivors compared to 8.9 mg/dL in those who died (*p* < 0.0001) [70], while the study conducted by Haltmeier et al. on trauma ICU patients demonstrated that TTR at admission was strongly correlated with infection rates (44.7% vs. 19% for values below or above 19 mg/dL, *p* < 0.001), mortality (17% vs. 7.4%, respectively, for the same cutoff, *p* < 0.007), and ICU length of stay (15.6 days vs. 7.8 days, respectively) [75]. In conclusion, changes in TTR over time are useful for understanding the patient’s clinical trajectory but also serve as an intermediate-term marker reflecting the inflammatory response and nutritional status in critically ill patients. This can help raise suspicion for diagnostic or therapeutic interventions when needed.

### Emerging Biomarkers and Future Perspectives for Nutritional Assessment: Interleukin-6

Interleukin-6 (IL-6) is a pro-inflammatory cytokine that plays a critical role in the immune response. Beyond its role in systemic inflammation, IL-6 has recently gained attention in nutritional assessment, particularly as a component of the NUTRIC (Nutrition Risk in Critically ill) score, which helps identify ICU patients at risk of malnutrition and guides nutritional therapy. The NUTRIC score, developed to assess nutritional risk in critically ill patients, incorporates IL-6 levels to reflect the inflammatory burden, which is strongly linked to hypermetabolism and muscle wasting. Patients with high NUTRIC scores (and elevated IL-6) are considered high-risk and may benefit more from early and aggressive nutritional interventions. This approach helps personalize nutritional strategies, aiming to mitigate the detrimental effects of systemic inflammation and improve recovery [76].

## 9. Clinical Tips: Practical Use of Biomarkers in the ICU

In the intensive care unit, selecting the right biomarkers and using them effectively can significantly impact patient outcomes. Each biomarker has its strengths and limitations, and understanding these is essential for accurate diagnosis and management. The effective use of biomarkers in the ICU, as reported in Figure 3, provides clinicians with actionable insights to improve patient management. NSE is highly reliable for assessing neurological outcomes after cardiac arrest, particularly when combined with tools such as SSEPs, EEG, and neuroimaging [77]. Its multimodal application ensures accurate prognostication while mitigating the risk of misinterpretation. Similarly, PCT aids in identifying bacterial infections and guiding antibiotic therapy, reducing unnecessary antibiotic exposure when integrated into clinical algorithms [78]. NT-proBNP is an essential biomarker for diagnosing heart failure and distinguishing cardiac from pulmonary causes of dyspnea, especially in emergencies [79]. For renal function, CysC outperforms SCr in detecting early AKI, making it a go-to biomarker for critically ill patients [80]. Moreover, IL-6 and TTR provide key information on systemic inflammation and nutritional status, respectively, supporting tailored interventions [32]. However, certain limitations require careful consideration. SCr, though widely used, is a slower and less reliable marker for early AKI, particularly in ICU settings where confounding factors like muscle mass variations can distort results. NSE, while valuable, must be used alongside other diagnostics due to potential confounders like hemolysis [81]. Likewise, PCT and IL-6 require contextual interpretation to avoid misdiagnosis in non-bacterial or systemic inflammatory conditions [82]. Finally, TTR should not be used as a standalone nutritional marker but as part of a comprehensive evaluation reflecting metabolic and inflammatory states.

## 10. Future Advances in Biomarker Utilization in the ICU

The future of biomarkers in the ICU is poised to transform patient care, enabling more precise diagnosis, monitoring, and treatment. Multiplex panels that combine biomarkers like NSE, IL-6, and PCT, paired with Artificial Intelligence [83] will enhance diagnostic accuracy and predict outcomes in conditions like sepsis and ARDS. Similarly, point-of-care testing devices for biomarkers such as CysC or NT-proBNP will provide rapid, bedside results, allowing clinicians to make timely decisions. Biomarkers are also paving the way for precision medicine, where stratifying patients based on markers like IL-6 can tailor therapies, such as immunomodulators for cytokine storms or antibiotics for bacterial infections [84]. Emerging markets like NfL for neuronal injury and KIM-1 for renal dysfunction are set to offer greater specificity and sensitivity, improving early diagnosis and intervention. Advances in omics technologies will further personalize care by identifying molecular profiles to guide real-time treatment adjustments [85]. Efforts to standardize biomarker assays and make testing more accessible will expand their use globally. Beyond acute care, biomarkers like TTR and NT-proBNP show promise in predicting long-term complications, helping guide recovery strategies after ICU discharge [86,87]. These advancements will shift biomarkers from diagnostic aids to central components of personalized ICU care, equipping clinicians to deliver targeted, timely, and effective interventions. Finally, it is important to discuss point-of-care testing (POCT) as an emerging strategy in critical care. POCT enables rapid bedside assessment of biomarkers, facilitating real-time decision-making and ultimately improving patient outcomes. In the ICU setting, where timely intervention is crucial, POCT for biomarkers such as BNP, Endothelin-1, IL-6, NGAL, and PCT offers significant advantages over traditional laboratory testing. Integrating POCT into ICU protocols can significantly enhance timely clinical decision-making, reduce diagnostic delays, and optimize patient management. While not all biomarkers currently have POCT options available, technological advancements are expanding their feasibility. The future of ICU biomarker analysis will likely rely on a combination of POCT, continuous monitoring systems, and artificial intelligence-driven interpretation to further improve critical care outcomes.

## 11. Conclusions

Integrating biomarkers into a broader diagnostic and therapeutic framework, clinicians can make more informed decisions. The key lies in using them contextually acknowledging their strengths while being mindful of their limitations. When interpreted thoughtfully, these tools enhance diagnostic precision, guide targeted interventions, and ultimately improve patient care. The future of biomarkers in the ICU is both exciting and transformative. With advances in technology, analytics, and personalized care, biomarkers will become more than diagnostic tools—they will serve as integral components of dynamic, patient-centered treatment strategies. By continuing to refine biomarker utility and integrating them into a multimodal framework, clinicians will be better equipped to meet the challenges of critical care and improve outcomes for their patients.

## Figures and Tables

**Figure 1 diagnostics-15-00975-f001:**
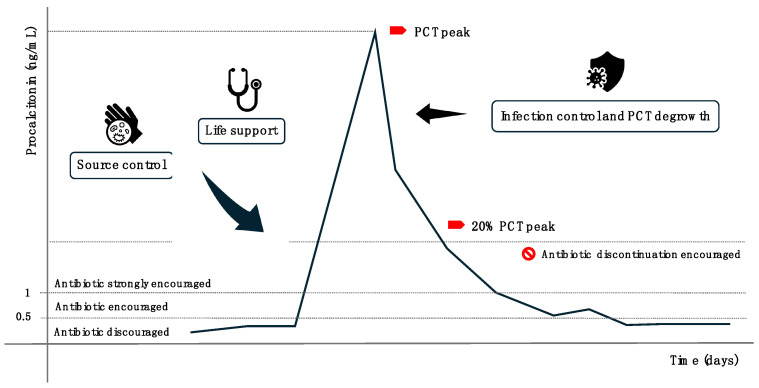
Schematic representation of the theoretical use of PCT in guiding antibiotic therapy during clinical practice. A PCT level above 0.5–1.0 suggests initiating antibiotic treatment, while a decrease below 20% of the peak value indicates consideration for discontinuation Arrows illustrate the clinical approach and the resulting progression of the infection over time.

**Figure 2 diagnostics-15-00975-f002:**
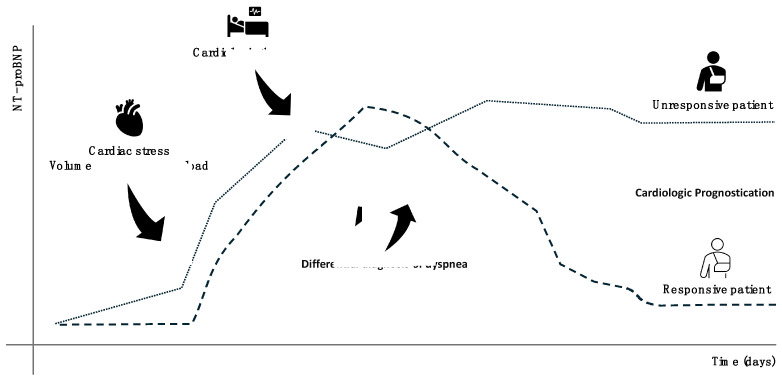
Schematic representation of the theoretical use of NT-proBNP for heart failure detection and monitoring the response to cardiologic therapy. Arrows highlight clinical events corresponding to fluctuations in NT-proBNP levels over time.

**Figure 3 diagnostics-15-00975-f003:**
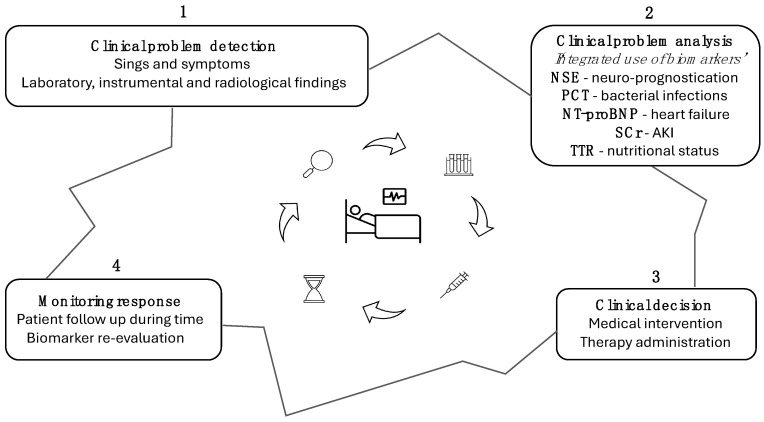
Flowchart: integrating biomarkers into clinical practice. Arrows highlight the cyclical pattern of problem detection, intervention, and subsequent re-evaluation in clinical practice.

**Table 1 diagnostics-15-00975-t001:** Characteristics of NSE: a biomarker for neurological damage and dysfunction.

Condition	Biological Mechanism	Cut-Off NSE	Sensitivity (%)	Specificity (%)	Main Findings
Out-of-Hospital Cardiac Arrest (OHCA)	Cerebral ischemia and reperfusion injury.	>20 μg/L (Days 3–4)	85	82	- NSE > 30 μg/L (Day 4) → 100% specificity for poor prognosis. - 48:24 h NSE ratio ≥1.7 → 100% specificity. [3]
Sepsis-Associated Encephalopathy (SAE)	Neuronal damage due to inflammation and BBB disruption.	14.36 μg/L (Day 3)	61.1	73.9	- NSE elevated in SAE vs. controls (Δ 7.79 ng/mL, 95% CI: 5.23–10.34). - Moderate diagnostic accuracy, improves with IL-6 [6].
Status Epilepticus (SE)	Neuronal damage from prolonged seizures.	17.8 μg/L	77.3	45.2	- NSE alone: limited accuracy (AUC = 0.624). - With S100β → AUC = 0.748, better specificity [2].
Delirium in the ICU	Acute brain dysfunction, unclear NSE role.	Not defined	NA	NA	- NSE is not a direct mortality predictor. - May indicate disease severity/ventilation need [12].
Ischemic/Traumatic Brain Injury	Neuronal apoptosis and necrosis.	>25 μg/L	76	80	- NSE correlates with injury severity. - Specificity affected by sepsis, shock [1].

**Table 2 diagnostics-15-00975-t002:** Comparison between SCr and CysC.

Characteristic	Serum Creatinine (SCr)	Cystatin C (CysC)
Production	Derived from muscle metabolism	Produced constantly by all nucleated cells
Elimination	Filtered by glomeruli; partially secreted by tubules	Filtered and fully metabolized in the proximal tubules
Influencing Factors	Muscle mass, diet, age, and hydration	Minimal; thyroid and inflammation influence
Cost	Low (<€5)	High (~10 × SCr)
Half-Life	~4 h	~1.5–2 h; faster response to GFR changes
Sensitivity for AKI	Low; delayed detection	High; AUROC 0.89 for AKI
Specificity for CKD	Moderate	High; better predictor of CKD progression
Response to Therapy	Slow; lag in reflecting renal recovery	Fast; better indicator of therapy response
Utility in Pediatric Patients	Limited due to growth-related variability	Effective; reliable, even in pediatric populations
Utility in Post-Transplant Monitoring	Limited; less accurate for dynamic GFR changes	Superior marker for post-transplant renal function monitoring
Utility in Critical Care	Limited in ICU; confounded by muscle wasting	High; preferred in ICU settings for early AKI detection
Correlation with Inflammation or Other Conditions	Minimal inflammation impact	May be influenced by systemic inflammation
Primary Applications	CKD monitoring, basic renal evaluation	Early AKI detection, ICU, cardiovascular risk prediction

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
