# Peer review of "ICU ‘Magic Numbers’: The Role of Biomarkers in Supporting Clinical Decision-Making"

_diagnostics, 2025, doi:10.3390/diagnostics15080975_

Round 1

Reviewer 1 Report

Comments and Suggestions for Authors

This is an incredible review of the common biomarkers currently used in the ICU. I was impressed with the detail provided by the authors. I believe the selected biomarkers are very relevant to intensivists. The table at the end, highlighting "recommended use" and when to "use with caution," provides an excellent summary of the review. The references are also impressive. I rarely suggest that a manuscript be considered for early publication, but I do not have much to suggest otherwise at this point. This will likely be of great interest to any critical care readership. It is very well written. The graphs are helpful in understanding the peak/plateau levels of procalcitonin and pro-BNP in Figures 1 and 2, respectively. Table 2, which compares serum Cr and Cystatin C, also serves as a strong encouragement for intensivists to consider more frequent use of Cystatin C. Cystatin C is still underutilized and has the potential to significantly enhance the clinical care of ICU patients.

Reviewer 1 Report

Comment 1: This is an incredible review of the common biomarkers currently used in the ICU. I was impressed with the detail provided by the authors. I believe the selected biomarkers are very relevant to intensivists. The table at the end, highlighting "recommended use" and when to "use with caution," provides an excellent summary of the review. The references are also impressive. I rarely suggest that a manuscript be considered for early publication, but I do not have much to suggest otherwise at this point. This will likely be of great interest to any critical care readership. It is very well written. The graphs clarify the peak/plateau levels of procalcitonin and pro-BNP in Figures 1 and 2, respectively. Table 2, which compares serum Cr and Cystatin C, also serves as a strong encouragement for intensivists to consider more frequent use of Cystatin C. Cystatin C is still underutilized and has the potential to significantly enhance the clinical care of ICU patients.

Response:    We are sincerely grateful for the reviewer’s generous and thoughtful feedback. It is truly rewarding to know that our work was perceived as both comprehensive and clinically relevant. We are especially humbled by the reviewer’s suggestion for early publication—an endorsement we regard as both motivating and deeply validating. We sincerely thank the reviewer for their high praise and support of our manuscript.

Reviewer 2 Report

Comments and Suggestions for Authors

Congratulations on your great work reviewing the roles of biomarkers in critical care. The authors reviewed various biomarkers related to ICU care, including NSE, procalcitonin, NT-proBNP, etc. This manuscript can be helpful in guiding the use of biomarkers in the ICU. Despite being a good review article, I think this manuscript can be revised.

Major Concerns

  1. Procalcitonin (PCT) is a biomarker to detect infection and guide antibiotic treatment.
    Despite the usefulness of PCT, the role of Presepsin has recently been reported in infection diagnosis. Can you add a short review about Presepsin?

  2. The role of NT-proBNP (BNP) in heart failure.
    BNP is a diagnostic marker for heart failure. The authors describe the role of BNP in heart failure. However, BNP can be elevated during fluid therapy in critically ill patients, so it is also used to evaluate volume overloading. Moreover, BNP is elevated in cases of septic cardiomyopathy or stress-induced cardiomyopathy (SCMP). But to diagnose these conditions, a combination of multiple cardiac biomarkers and echocardiography is mandatory. The authors just focused on heart failure, but BNP monitoring is not only for heart failure. Please add the role of BNP in diagnosing and guiding cardiomyopathy and fluid therapy.

  3. Role of Endothelin
    Endothelin-1 is a potent vasopressor and a marker of endothelial dysfunction. I think adding endothelin would be helpful in guiding the management of ICU patients. BNP and endothelin can also be described in the same section.
    The section titled “N-terminal pro-brain natriuretic peptide (NT-proBNP): A Reliable Biomarker for Cardiac Failure Diagnosis and Management” can be renamed as “Markers of Cardiovascular Dysfunction: NT-proBNP and Endothelin-1.”

  4. Role of IL-6
    Recently, IL-6 has been used as a component of the NUTRIC score to screen ICU malnutrition and guide nutritional therapy. Please include the role of IL-6 in nutritional assessment.

  5. Essential biomarkers for renal function
    The authors suggested SCr and CysC as biomarkers for renal function. However, in AKI patients, NGAL is a more specific and rapid diagnostic biomarker. The role of NGAL should be addressed.

  6. Lactate
    Lactate is a simple biomarker to diagnose and monitor tissue perfusion. Can you add a section for lactate?

Minor Concerns

  1. Legends of tables and captions of figures are not addressed.
    They should be included to explain the results or flow of the figures.

  2. Despite the good explanation of the roles of biomarkers, can you add the possibility of POCT in the ICU?

  3. Can you provide a simple diagram or image to support clinical tips on the practical use of biomarkers (instead of a table or a new image)?

Reviewer 2 Report

Comment 1:   Congratulations on your great work reviewing the roles of biomarkers in critical care. The authors reviewed various biomarkers related to ICU care, including NSE, procalcitonin, NT-proBNP, etc. This manuscript can be helpful in guiding the use of biomarkers in the ICU. Despite being a good review article, I think this manuscript can be revised.

Response:   We are truly grateful to the reviewer for the kind words and for recognizing the potential value of our review in supporting biomarker use in the ICU. It is an honor to receive such positive feedback from an expert in the field. We fully acknowledge and appreciate the suggestion for further revision, and we have carefully reviewed the manuscript to improve clarity, strengthen the structure, and refine several key sections. We hope the updated version better meets the standards expected for publication. We sincerely thank the reviewer again for their thoughtful input and encouragement.

Comment 2: Procalcitonin (PCT) is a biomarker to detect infection and guide antibiotic treatment. Despite the usefulness of PCT, the role of Presepsin has recently been reported in infection diagnosis. Can you add a short review about Presepsin?

Response:  thank you for the valuable suggestion. In response, we have included a brief review titled ‘Emerging Biomarkers and Future Perspectives for Sepsis Detection: Presepsin’ at the end of the paragraph on Procalcitonin.

Comment 3:  The role of NT-proBNP (BNP) in heart failure. BNP is a diagnostic marker for heart failure. The authors describe the role of BNP in heart failure. However, BNP can be elevated during fluid therapy in critically ill patients, so it is also used to evaluate volume overloading. Moreover, BNP is elevated in cases of septic cardiomyopathy or stress-induced cardiomyopathy (SCMP). But to diagnose these conditions, a combination of multiple cardiac biomarkers and echocardiography is mandatory. The authors just focused on heart failure, but BNP monitoring is not only for heart failure. Please add the role of BNP in diagnosing and guiding cardiomyopathy and fluid therapy.

Response:  Following your suggestion we have added a brief paragraph discussing the use of NT-proBNP in multiple clinical contexts, such as perioperative risk assessment, diagnosis of pulmonary hypertension, evaluation of volume overload, assessment of septic and stress-induced cardiomyopathy

Comment 4:  Role of Endothelin. Endothelin-1 is a potent vasopressor and a marker of endothelial dysfunction. I think adding endothelin would be helpful in guiding the management of ICU patients. BNP and endothelin can also be described in the same section. The section titled “N-terminal pro-brain natriuretic peptide (NT-proBNP): A Reliable Biomarker for Cardiac Failure Diagnosis and Management” can be renamed as “Markers of Cardiovascular Dysfunction: NT-proBNP and Endothelin-1

Response:  We have included a dedicated paragraph titled ‘Emerging Biomarkers and Future Perspectives for Heart Failure: Endothelin-1,’ which provides a brief review of endothelin-1 in the context of endothelial dysfunction and heart failure evaluation.

Comment 5:  Role of IL-6. Recently, IL-6 has been used as a component of the NUTRIC score to screen ICU malnutrition and guide nutritional therapy. Please include the role of IL-6 in nutritional assessment

Response: In response to your suggestion, we have added a dedicated paragraph titled ‘Emerging Biomarkers and Future Perspectives for Nutritional Assessment: Interleukin-6,’ which discusses the role of interleukin-6 as a component of NUTRIC. This paragraph is placed at the end of the section on Prealbumin.

Comment 6: Essential biomarkers for renal function. The authors suggested SCr and CysC as biomarkers for renal function. However, in AKI patients, NGAL is a more specific and rapid diagnostic biomarker. The role of NGAL should be addressed.

Response:  We sincerely thank the reviewer for this important observation. We fully acknowledge the growing interest in NGAL (neutrophil gelatinase-associated lipocalin) as an early and sensitive biomarker for acute kidney injury (AKI), particularly due to its ability to detect structural tubular damage before changes in serum creatinine become evident. However, our review specifically aimed to focus on biomarkers of renal function rather than structural injury. SCr and Cystatin C remain the most widely used markers for assessing glomerular filtration in daily ICU practice, and they are more readily available across most clinical settings. While NGAL certainly shows promise, it also presents several limitations that have hindered its routine adoption: 1)Limited availability in many institutions and ICU settings; 2)Lack of standardization across assays and reference ranges;3) Low specificity, as levels can also be elevated in non-renal inflammatory states, such as sepsis, trauma, or malignancy; 4) Cost considerations, which may limit its use in resource-constrained environments. Considering these factors, we chose to prioritize functional biomarkers that are both validated and more accessible in critical care settings. Nonetheless, we have now added a brief mention of NGAL in the revised manuscript to acknowledge its role and highlight the evolving landscape of renal biomarkers.

Comment 7. Lactate. Lactate is a simple biomarker to diagnose and monitor tissue perfusion. Can you add a section for lactate?

Response:  We thank the reviewer for this pertinent comment. We fully agree that lactate is a well-established and widely used biomarker for assessing tissue perfusion and guiding resuscitation in critically ill patients. Its role in the early identification of shock and in monitoring therapeutic response is undoubtedly essential in ICU practice. However, in this review, our goal was to focus primarily on organ-specific biomarkers that directly reflect the function or pathology of individual systems, such as cardiac, renal, neurological, and hepatic biomarkers. Lactate, while critically important, is a global marker of hypoperfusion and metabolic stress, rather than being specific to a single organ or pathological process. To maintain a focused and coherent scope, we therefore chose not to include lactate in the current manuscript.

Comment 8. Minor Concerns.

  • Legends of tables and captions of figures are not addressed. They should be included to explain the results or flow of the figures.

Response: The legends have been updated and are now addressed in the revised manuscript.

  • Despite the good explanation of the roles of biomarkers, can you add the possibility of POCT in the ICU?

Response: We have discussed the role of POCT in the concluding paragraph of the revised manuscript.

  • Can you provide a simple diagram or image to support clinical tips on the practical use of biomarkers (instead of a table or a new image)?

Response: In response to your suggestion, we have replaced Table 3 with a figure illustrating the theoretical flowchart of biomarker integration in clinical practice.

Round 2

Reviewer 2 Report

Comments and Suggestions for Authors

Well addressed to comments